# Ecological correlates to cranial morphology in Leporids (Mammalia, Lagomorpha)

Brian P. Kraatz[1], Emma Sherratt[2], Nicholas Bumacod[3] and Mathew J. Wedel[1,4]

[1] College of Osteopathic Medicine of the Pacific, Western University of Health Sciences, Pomona, CA, USA
[2] Department of Ecology, Evolution, and Organismal Biology, Iowa State University, Ames, IA, USA
[3] College of Dental Medicine, Western University of Health Sciences, Pomona, CA, USA
[4] College of Podiatric Medicine, Western University of Health Sciences, Pomona, CA, USA

## ABSTRACT

The mammalian order Lagomorpha has been the subject of many morphometric studies aimed at understanding the relationship between form and function as it relates to locomotion, primarily in postcranial morphology. The leporid cranial skeleton, however, may also reveal information about their ecology, particularly locomotion and vision. Here we investigate the relationship between cranial shape and the degree of facial tilt with locomotion (cursoriality, saltation, and burrowing) within crown leporids. Our results suggest that facial tilt is more pronounced in cursors and saltators compared to generalists, and that increasing facial tilt may be driven by a need for expanded visual fields. Our phylogenetically informed analyses indicate that burrowing behavior, facial tilt, and locomotor behavior do not predict cranial shape. However, we find that variables such as bullae size, size of the splenius capitus fossa, and overall rostral dimensions are important components for understanding the cranial variation in leporids.

## INTRODUCTION

The relationship between form and function as it relates to locomotion has been extensively studied in a wide range of vertebrate groups (*Webb, 1984*; *Hildebrand, 1988*; *Rayner, 1988*; *Aerts et al., 2000*). The mammalian order Lagomorpha has been the subject of many morphometric studies aimed at understanding this relationship in postcranial morphology (e.g., *Reese, Lanier & Sargis, 2013*; *Fostowicz-Frelik, 2007*; *Seckel & Janis, 2008*; *Young et al., 2014*), and the impetus of these is largely to understand the high-speed form of leaping observed in some leporids (rabbits and hares). Leporids are peerless cursors for their size; some hares have been shown to achieve speeds greater than 70 km/h (*Garland, 1983*). Indeed, the leporid postcranial skeleton exhibits many derived features that are strongly associated with saltation and cursoriality, including limb element elongation (*Szalay, 1985*; *Fostowicz-Frelik, 2007*; *Seckel & Janis, 2008*).

Corresponding author
Brian P. Kraatz,
bkraatz@westernu.edu

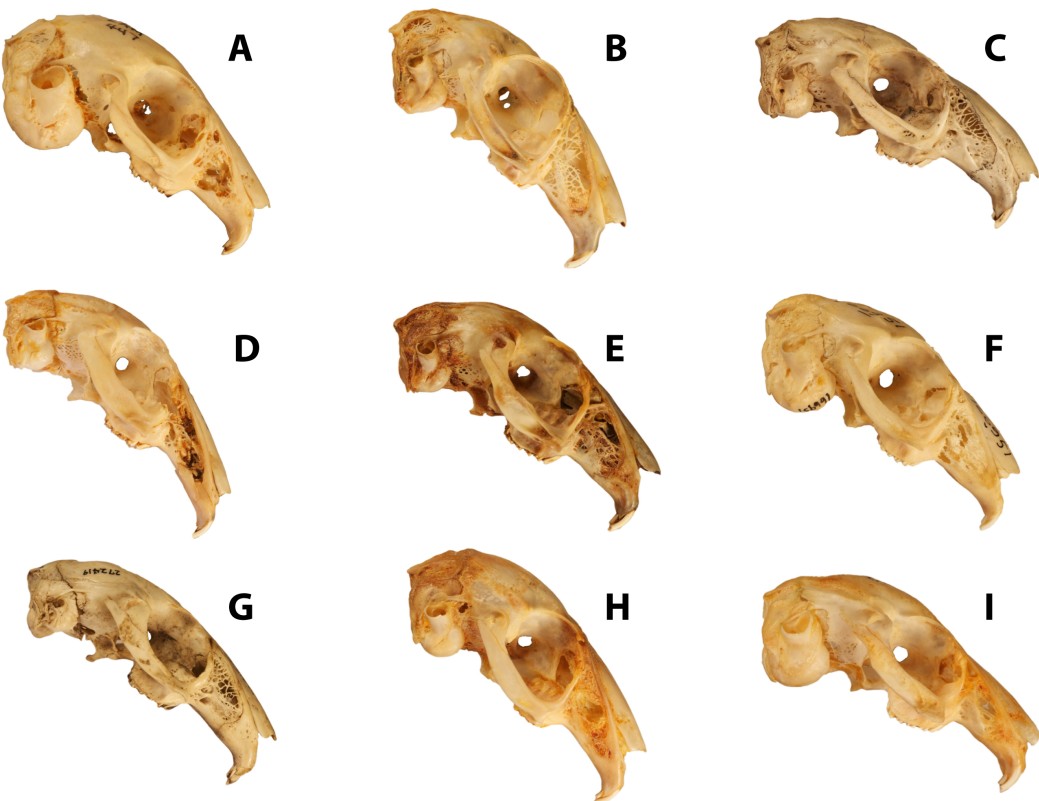

**Figure 1 Disparity of leporid skulls.** Disparity in facial tilt and cranial shape in selected leporids, including (A) *Brachylagus idahoensis* (LACM 447; SLD ∼50 mm), (B) *Lepus capensis* (LACM 40152; SLD ∼82 mm), (C) *Poelagus marjorita* (AMNH 51056; SLD ∼80 mm), (D) *Pronolagus crassicaudatus* (AMNH 89033; SLD ∼80 mm), (E) *Lepus americanus* (LACM 93737; SLD ∼75 mm), (F) *Oryctolagus cuniculus* (AMNH 166951; SLD ∼78 mm), (G) *Nesolagus timminsi* (AMNH 272419; SLD ∼78 mm), (H) *Bunolagus monticularis* (AMNH 146662; SLD ∼78 mm), and (I) *Romerolagus diazi* (AMNH 148181; SLD ∼60 mm). All skull images are scaled to approximately the same skull length, skull length measurements are an approximation based on our measured specimens.

The cranial skeleton is more often overlooked in studies of form and locomotion, though there are biologically relevant associations between skull form and locomotor behavior, such as the role of the skull in active headfirst burrowing (e.g., *Gans, 1974*; *Barros, Herrel & Kohlsdorf, 2011*; *Sherratt et al., 2014*; *Hopkins & Davis, 2009*; and see *Wake, 1993* for a review). In leporids, it has been suggested that morphological transformations of the skull may also be related to their ecology, particularly locomotion and vision (*DuBrul, 1950*; *Bramble, 1989*). The leporid skull is highly transformed, exhibiting a combination of features that clearly distinguish it from a more typical mammalian skull. A striking, yet often overlooked, characteristic is the broad dorsal arching of the cranium (*Thompson, 1942*), which is achieved via expansion and folding of the supraoccipital, and a distinct flexure near the basisphenoid/presphenoid suture (Fig. 1). A prominent ridge on the dorsal portion of the posterior cranial roof, which is superficially similar to an occipital crest,

**Peer**J ________________________________

**Table 1 Leporid species studied.** See Appendix S1 for specific specimens measured, and the text for discussion regarding the assessment of ecological variables.

| Species | Locomotion type | Burrowing | Abbreviation | *n* |
|---|---|---|---|---|
| *Romerolagus* | Saltatorial | Yes | Ro | 7 |
| *Bunolagus* | Saltatorial | Yes | Bu | 2 |
| *Caprolagus* | Generalized | Yes | Ca | 2 |
| *Brachylagus* | Generalized | Yes | Br | 10 |
| *Sylvilagus floridanus* | Saltatorial | No | Sfl | 10 |
| *Sylvilagus palustris* | Generalized | No | Spal | 10 |
| *Sylvilagus audobonii* | Saltatorial | Yes | Sau | 10 |
| *Poelagus marjorita* | Saltatorial | No | Po | 10 |
| *Pronolagus crossicaudatus* | Saltatorial | No | Pc | 10 |
| *Oryctolagus cuninculus* | Saltatorial | Yes | Oc | 10 |
| *Nesolagus timminsi* | Saltatorial | Yes | Nt | 2 |
| *Lepus americanus* | Saltatorial | No | Lam | 10 |
| *Lepus timidus* | Saltatorial | Yes | Lti | 10 |
| *Lepus capensis* | Cursorial | Yes | Lcap | 10 |
| *Lepus californicus* | Cursorial | No | Lcal | 12 |
| *Lepus saxatilis* | Cursorial | No | Lsax | 9 |

is actually a distinct flexure within the supraoccipital bone. Based on the position of the rabbit skull in resting position (*De Beer, 1947*: Fig. 9; *Vidal, Graf & Berthoz, 1986*: Fig. 4B, and see our Fig. 2), this flexure results in significant tilting of the facial region ventrally relative to the basicranium, which we here refer to as Facial Tilt (FT). *DuBrul (1950)* discusses this feature in detail within hares, and points out that the facial tilt of leporids is likely related to their unique mode of locomotor behavior. *DuBrul (1950)* also discusses the similarities in leporid skull transformations to those of our own lineage; in our hominin relatives, increased basicranial flexion is associated with the onset of bipedal locomotion (*Strait & Ross, 1999*).

The goal of this study is to investigate the relationship between cranial shape and locomotion (cursoriality, saltation, and burrowing) within crown leporids. Our study is driven by hypotheses previously stated (*DuBrul, 1950*; *Bramble, 1989*) but never quantitatively tested. We use a large morphometric dataset spanning 16 phylogenetically constrained extant taxa (Table 1) to evaluate hypotheses about the relationship between skull shape and facial tilt with locomotor ecology.

## STUDY SYSTEM AND HYPOTHESES

The mammalian order Lagomorpha is composed of two families, Leporidae (rabbits and hares) and Ochotonidae (pikas). Ochotonids are represented by one living genus, *Ochotona*, which includes two North American and 28 Eurasian species (*Alves & Hackländer, 2008*). Leporids include 11 living genera with 62 species overall. The majority of species are found within two genera (*Alves & Hackländer, 2008*); *Lepus* (hares, 32 species) and *Sylvilagus* (a portion of rabbits, 17 species). Of the remaining nine genera, seven are monotypic, while two genera, *Nesolagus* and *Pronolagus*, only include two

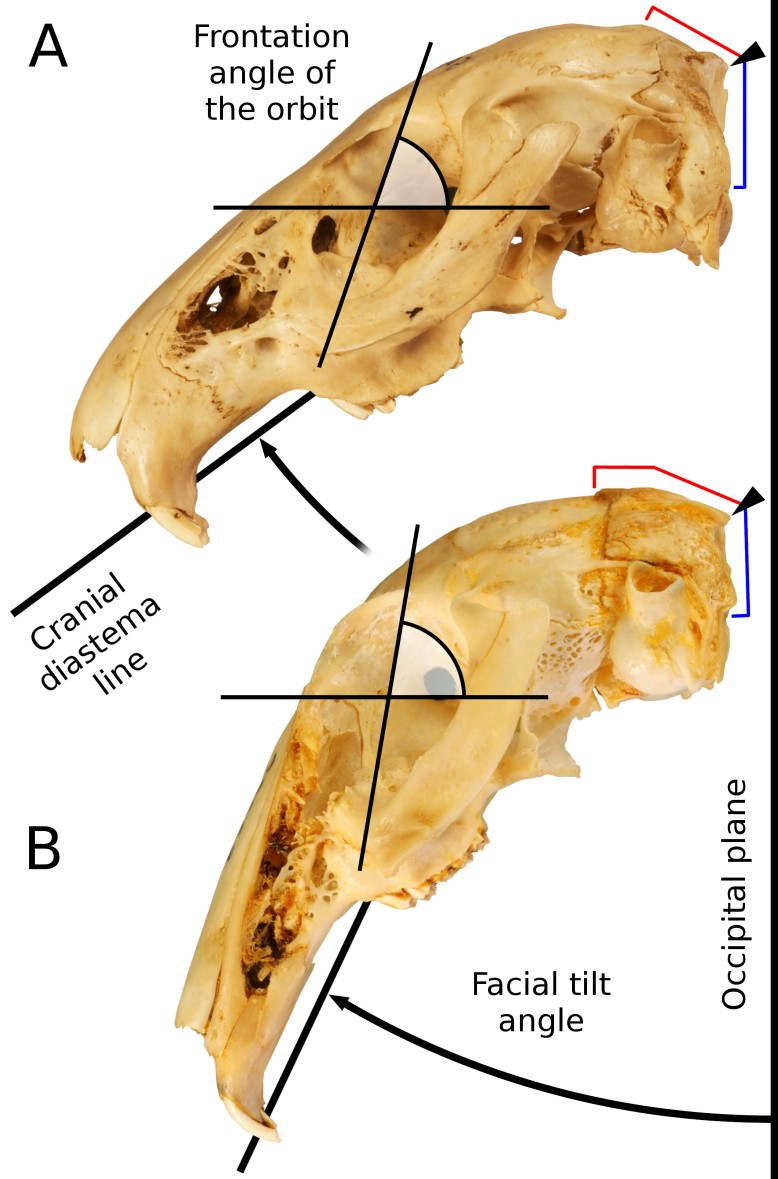

**Figure 2 Facial tilt in leporids.** The crania of (A) *Caprolagus hispidus* (AMNH 54852, above) and (B) *Pronolagus crassicaudatus* (AMNH 89033, below) are shown in left lateral view. Facial tilt (FT) is defined herein as the angle between the upper diastema and the occipital plane, where increased values indicated a skull orientation closer the horizontal plane. The triangle indicates the position of the external occipital protuberance (EOP), and from that, both the dorsal (red) and occipital (blue) extent of the supraoccipital bones is outlined.

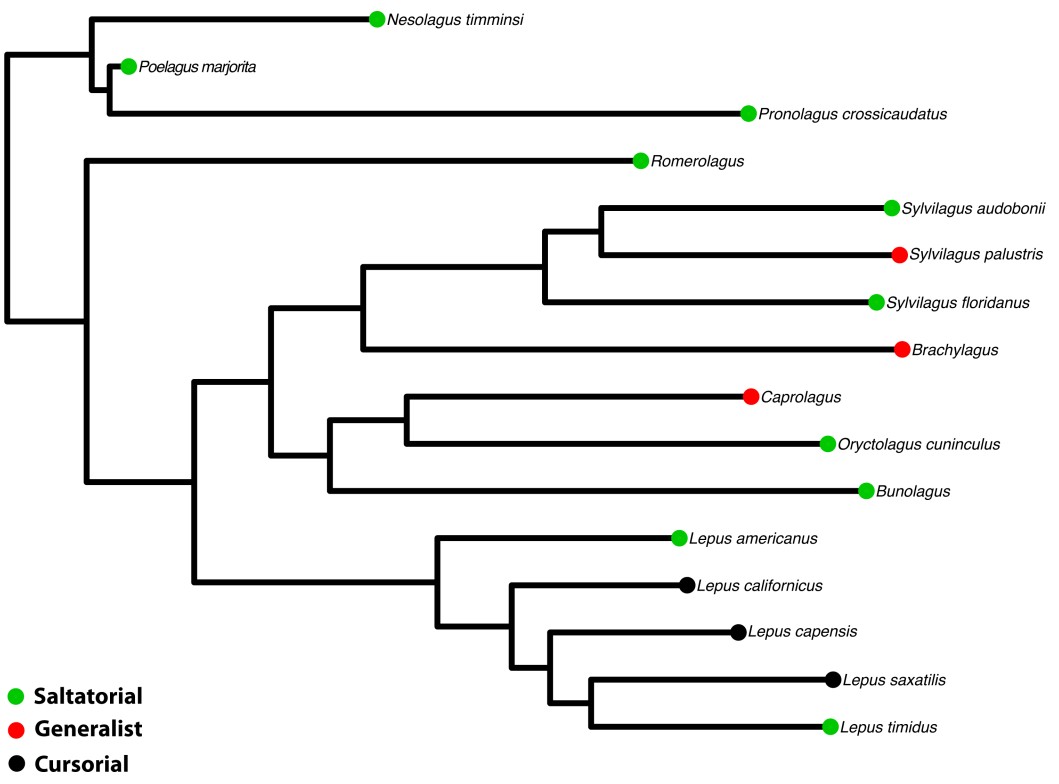

**Figure 3 Phylogeny of Leporidae.** The phylogenetic hypothesis of the 16 taxa used in this study, pruned from the supermatrix maximum likelihood phylogeny in *Matthee et al. (2004)*. Locomotor styles from Table 1.

and four species, respectively. Overall, sixteen leporids species are currently considered endangered or critically endangered by the IUCN (*Alves & Hackländer, 2008*), and conservation issues are compounded by the lack of natural history data for many of these species. Leporids are found on every continent except Antarctica, from the high arctic to dry, hot deserts (*Chapman & Flux, 1990*; *Chapman & Flux , 2008*). Some leporids are nocturnal, some are social, and some live in dense cover as opposed to the open plains often associated with these animals (*Stoner, Bininda-Emonds & Caro, 2003*). In terms of size, our study includes (Appendix S1) the smallest leporid, *Brachylagus* (mean skull length ∼50 mm) to one of the largest, *Lepus timidus* (mean skull length ∼90 mm). Genera such as *Pentalagus* and *Caprolagus* have heavy, robust skulls, compared to the typically gracile skulls of most taxa. These cranial differences manifest themselves morphometrically via a wide range of snout lengths and marked differences in skull robustness and form (Fig. 1). While leaping abilities are common among most leporid lineages, they are also known to be facultatively semiaquatic, scansorial, fossorial, or exhibit a more generalized, non-hopping form of locomotion (*Chapman & Flux, 1990*). We distinguish here between the saltatory locomotion (i.e., hopping) most typical among leporids (Table 1 and Fig. 3), and its cursorial form observed in some hare lineages (*Gambaryan & Hardin, 1974*; *Bramble, 1989*). Generalists are recognized as those who don't exhibit clear hopping, but rather move in a more scampering habit.

### Hypothesis 1–facial tilt

***A high degree of facial tilting (e.g., ventral flexion of the facial region) should (a) be positively correlated with more active (e.g., saltatorial or cursorial) locomotor styles, and (b) show no correlation with burrowing habit***

Variation in the degree of facial tilt among leporids has strong effects on orbital orientation (Fig. 3). There is substantial literature discussing the relationship between orbit orientation and ecology within vertebrates (*Noble, Kowalski & Ravosa, 2000*; *Cox, 2008*; *Heesy, 2008*; *Iwaniuk et al., 2008*; *Jeffery & Cox, 2010*), and *Cartmill (1970)* established the terms 'orbital convergence' and 'frontation' to understand these relationships. While orbit orientation is influenced by brain size and jaw mastication (*Lieberman, Ross & Ravosa, 2000*; *Cox, 2008*), within primates, orbital convergence is also strongly associated with increased binocular visual field overlap observed in nocturnal predatory species (*Heesy, 2004*; *Heesy, 2008*). Various groups exhibit a high degree of both orbital convergence and orbital frontation (*Cox & Jeffery , 2008*), with hominids serving as an exemplar; orbital frontation is strongly positively correlated with basicranial flexion (*Ross, 1995*). As *DuBrul (1950)* points out, facial tilt transformations among leporids are nearly identical to basicranial flexion observed within anthropoids; increased facial tilt and basicranial flexion both result in increased orbital frontation (see Fig. 2 for changes in frontation related to increased FT). Several workers have shown that increased frontation is positively correlated with arboreal taxa (*Cartmill, 1970*; *Heesy, 2008*); increased frontation changes the visual field to allow for better visualization of substrate. *Jeffery & Cox (2010)* show that leporids have relatively low degrees of convergence and frontation. As we discuss below, however, when facial tilt is taken into consideration, leporids actually demonstrate a relatively higher degree of frontation (as indicated by the orbital plane relative to the vertical plane). More importantly, regardless of the absolute measure of frontation within leporids, we expect that frontation will vary among leporids correlated with varying degrees of facial tilt. For this reason, we expect that facial tilt (as a proxy for frontation) should be strongly correlated to locomotor styles that would require enhanced substrate perception (saltatorial and cursorial), but we do not expect that facial tilt will be related to burrowing habit.

### Hypothesis 2—Skull shape

***We expect that there will be significant skull shape differences among (a) locomotor styles, and (b) burrowing habits***

We have no *a priori* expectations about how overall skull shape might change with locomotor mode or burrowing habit. Instead we will investigate the more fundamental question of whether skull shape is related to locomotion and burrowing habit at all. Our interest in this question is therefore more a form of exploratory data analysis than a test of a specific hypothesis.

## MATERIALS AND METHODS

We collected morphometric data (Table 2 and Appendix S1) from 140 leporid skulls spanning 16 taxa (Table 1) housed in the departments of Mammalogy at the American

Table 2 **Skull measurements used.** Variables used in this study and description; see Figs. 2 and 4 for illustrations of the measurement conventions.

| Abbr. | Variable | Measurement convention |
|---|---|---|
| BLD | Bulla diameter | Maximum diameter (in any direction) of right bulla |
| BOL | Basioccipital length | Maximum midsagittal length from anterior basioccipital to foramen magnum |
| DIL | Diastema length | Maximum distance between right I2 and M1 |
| IOW | Interorbital width | Minimum transverse width between dorsal rims of orbits |
| NL | Nasal length | Maximum parasagittal length of nasal bones (i.e., orthogonal antero-posterior but not along midline) |
| NW | Nasal width | Maximum transverse width across posterior nasal bones |
| PAL | Parietal length | Maximum midsagittal length of parietal bones |
| SCF | Splenius capitis fossa | Maximum parasagittal length from anterior margin of *M. splenius capitis* insertion fossa to opisthocranion |
| SLD | Skull length dorsal | Maximum midsagittal length from anterior nasal bones to Opisthocranion (just dorsal to incisors) to opisthocranion |
| SW | Skull width | Maximum transverse width across zygomatic processes |

Table 3 **PCA loadings.** The first four principal component (PC) axes contribute to 90.2% of the total variation of the ten log normal variables. For each PC, the proportion of total variance (%) and the loadings on these are given. The variables with the highest loading are shown in bold and are discussed within the text.

| | PC1 | PC2 | PC3 | PC4 |
|---|---|---|---|---|
| Proportion of variance | 43.5 | 24.4 | 13.2 | 9.1 |
| BLD | **0.85046873** | 0.02545804 | 0.016857617 | 0.272291215 |
| BOL | 0.159862063 | 0.161106564 | 0.012761305 | −0.070830236 |
| DIL | −0.246528468 | −0.026035317 | **−0.415136231** | 0.156169866 |
| IOW | −0.249260494 | 0.331660879 | **0.681589555** | 0.259415969 |
| NL | −0.175108112 | 0.062932441 | **−0.505939657** | 0.130829626 |
| NW | −0.296951523 | 0.01926985 | 0.131337875 | 0.224903873 |
| PAL | 0.027064391 | 0.140782388 | 0.078947604 | **−0.866213361** |
| SCF | −0.050436754 | **−0.905402173** | 0.218089373 | −0.039704014 |
| SLD | −0.064043373 | 0.114113325 | −0.185926838 | 0.027064132 |
| SW | 0.044933539 | 0.076114004 | −0.032580602 | −0.093927069 |

Museum of Natural History (AMNH) and the Los Angeles County Museum of Natural History (LACM). Care was made to use only adult specimens, characterized by fully fused occipital sutures (*Hoffmeister & Zimmerman, 1967*). Ten linear measurements (Table 3 and Fig. 3) were recorded per specimen using digital calipers by three authors (BPK, MW, and NB), and a repeatability analysis (consisting of 10 specimens measured 3 times, results not presented) was performed to ensure there was no intercollector bias introduced. The ten cranial measurements were analyzed using the log-shape ratios approach (*Mosimann, 1970*; *Mosimann & James, 1979*). For each specimen, size was computed as the geometric mean of all measurements, and then each measurement was divided by size to obtain the shape ratios. We then used the log of this quantity as raw data for the subsequent analyses.

Facial tilt was measured by photographing each skull in lateral view using a Nikon D80 digital camera (Nikon, Tokyo, Japan). The skulls were placed in a

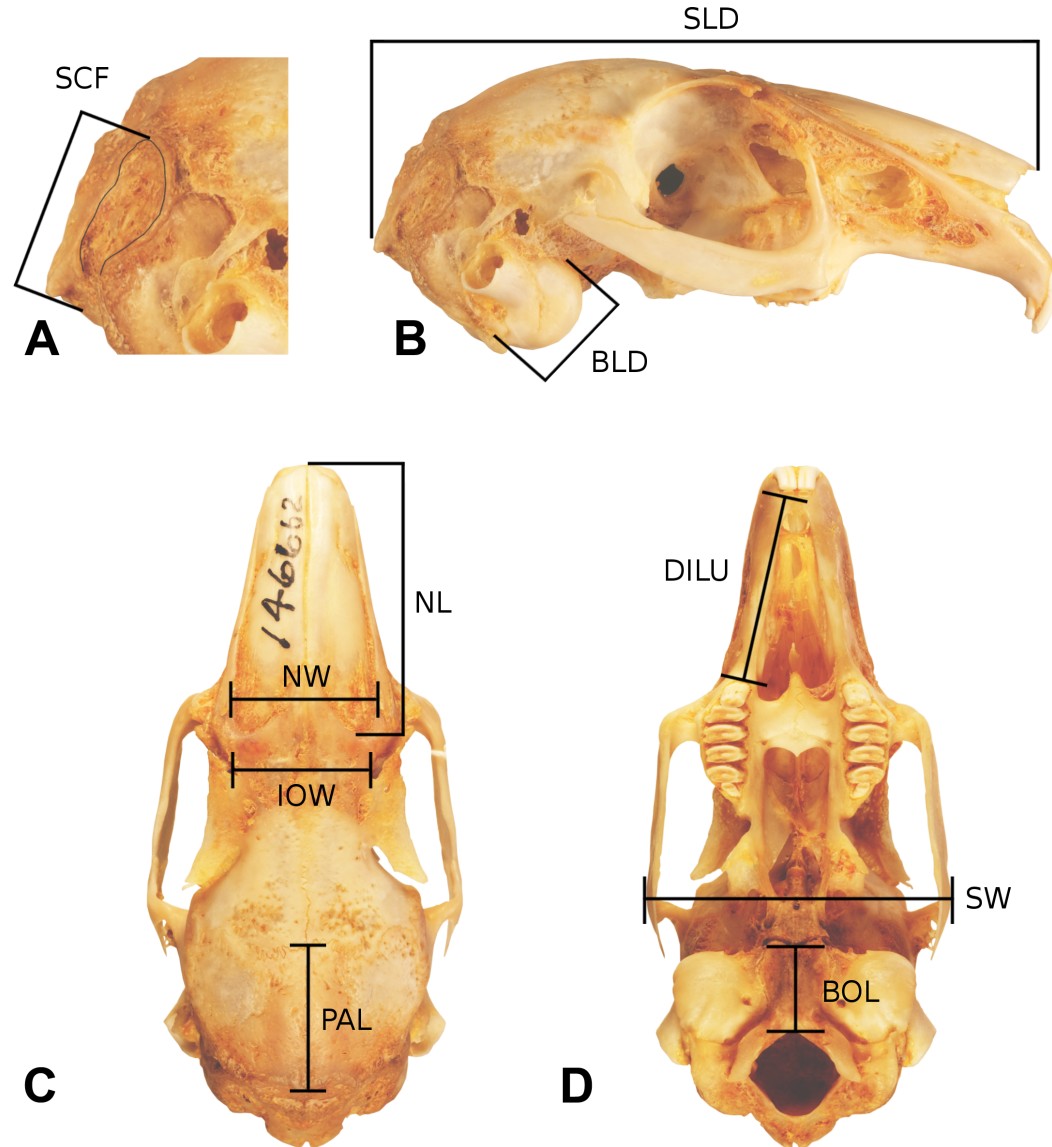

**Figure 4 Skull measurements.** A representative leporid skull showing measurements used in this analysis. The cranium of *Bunolagus monticularis* (AMNH 146662) is shown in ((A) and (B)) right lateral (top), (C) dorsal (lower left), and (D) ventral (lower right) views. Abbreviations follow Table 1. See figure 3 for a description of Facial Tilt (FT).

sandbox to ensure that the sagittal plane was orthogonal to the focal direction. Facial tilt angle was acquired from the digital photos within Adobe Photoshop©, measured as the angular difference between the 'occipital plane' and a line parallel to the cranial diastema (Fig. 3). Variation among individuals for the cranial variables weas explored using principal components analysis on the covariance matrix of the log-shape ratios shape variables within the statistical software R v3.1.1 (*R Core Team, 2014*, http://cran.r-project.org/).

## Phylogenetically informed analyses

To examine facial tilt angle and cranial shape in a phylogenetic context, we used the phylogenetic relationships among species of Leporidae recently published by *Matthee et al. (2004)*. The original tree was constructed using seven genes (five nuclear and 2 mt) for 25 ingroup taxa. We pruned the tree using Mesquite©(*Maddison & Maddison, 2015*) to include only the 16 taxa studied here (Fig. 3), and retained the information on branch lengths (details of which are in (*Matthee et al., 2004*)).

We first examined the amount of phylogenetic signal in the morphometric variables, calculating the *K* statistic (*Blomberg, Garland & Ives, 2003*) for facial tilt angle, and the multivariate equivalent $K_{mult}$ (*Adams, 2014a*) for all log-shape ratios. The *K* statistics provide a measure of the strength of phylogenetic signal for univariate and multivariate traits respectively, and in each case provides a single statistic. A value of less than one implies that taxa resemble each other phenotypically less than expected under Brownian motion, while values of more than 1 implies that close relatives are more similar to one another phenotypically than expected under Brownian motion. Significance testing was performed using a permutation procedure whereby the variables are randomized relative to the tree, and 1000 permutations were performed for each test (*Blomberg, Garland & Ives, 2003*).

Log-shape ratios and facial tilt angle were compared to several key ecological indicators, including locomotor type and burrowing habit (Table 1). Ecological data were obtained from *Chapman & Flux (1990)* and *Stoner, Bininda-Emonds & Caro (2003)*. We divided leporids into three locomotor categories: generalized or 'scramble' locomotors, which tend to be the slowest-moving; saltatory or hopping locomotors; and fast-moving taxa that practice cursorial (leaping and bounding) locomotion, which is essentially a specialized form of saltation. Regarding burrowing habits, some leporids dig their own burrows (e.g., *Oryctolagus* and *Romerolagus*), whereas others simply occupy preexisting burrows excavated by other animals. For the purposes of this study, we refer to leporids as burrowers if they occupy burrows consistently, regardless of whether they dig the burrows.

To test whether or not the degree of facial tilt differs among the three locomotor categories, we performed a one-way Analysis of Variance (ANOVA) in an evolutionary context, under a Brownian motion model of evolution. This was done by using species means of the FT angle in a distance-based phylogenetic generalized least squares analysis (D-PGLS; *Adams, 2014b*). A distance-based approach provides numerically identical estimates of evolutionary patterns to those obtained from standard implementations of PGLS on univariate datasets, and was used here for consistency with analyses below on the log-shape ratios. The statistical significance of each term in the D-PGLS was assessed using 1000 permutations whereby the species means are shuffled among the tips of the phylogeny. We performed a second ANOVA as above to test whether facial tilt differs between taxa that utilize burrows ("burrowing") and those that do not ("non-burrowing"). Box and whisker plots were used to visualize the individual variation in facial tilt angle among groups.

To test whether or not cranial shape, as represented by ten morphometric variables, differs among the three locomotor types, we performed a multivariate analysis of variance

**Peer**J

in an evolutionary context under a Brownian motion model of evolution. This was done as a D-PGLS with the species means of the ten log-shape ratios. The D-PGLS performs better than a regular PGLS when the number of variables begins to approach the number of species (*Adams, 2014b*). The statistical significance of each term in the D-PGLS was assessed using 1,000 permutations of the species means. Similarly, we tested whether or not cranial shape differs between burrowing and non-burrowing taxa using a D-PGLS as above.

Finally, to test whether or not facial tilt is a significant predictor of cranial shape, we performed a multivariate regression in an evolutionary context, under a Brownian motion model of evolution, again using the D-PGLS approach. The statistical significance was assessed using 1000 permutations of the species means of the log-shape ratios. All of the phylogenetically informed analyses were done using the geomorph package (*Adams et al., 2014*) in the statistical software R v3.1.1 (*R Core Team, 2014*). The ANOVAs on FT, the MANOVAs on cranial log-shape rations, and the multivariate regression were done using the *procD.pgls* function, and phylogenetic signal was calculated with the *physignal* function.

## RESULTS

### Facial tilt

Facial tilt (FT) summarizes the broad dorsal arching of the skull roof that is prominent among living leporids (Fig. 3). Across the species in this study, the measure of facial tilt angle has a very low value for $K$, implying that the taxa resemble each other morphologically less than expected under Brownian motion, and the test is not significant ($K = 0.62$, $P = 0.53$). Overall, there is a nearly 30° range of variation in FT among specimens of all species in this sample (Appendix S1). We found a significant difference among locomotor types for facial tilt angle (D-PGLS, $F = 7.02$, $P = 0.016$; Fig. 5A). The mean FT angle for generalized locomotors (mean, $\mu = 44.0$, standard deviation, $\sigma = 5.48$) is substantially higher than that of cursorial ($\mu = 36.3$, $\sigma = 5.46$) and saltatorial taxa ($\mu = 37.2$, $\sigma = 5.91$) (Fig. 5A). This indicates that taxa that are either saltatorial or cursorial tend to have facial regions that are more ventrally deflected. By contrast, we found no significant difference in FT angle between burrowing and non-burrowing taxa (Fig. 5B; D-PGLS, $F = 0.0037$, $P = 0.973$; Fig. 5B).

### Cranial shape analyses

In a principal components analysis of the ten log-shape ratios among individuals, the first four PC axes account for 90.2% of total variance. PC1 accounts for 43.5% of cranial shape differences, PC2 accounts for 24.4%, PC3 accounts for 13.2%, and PC4 accounts for 9.0% of variance. (Table 3 and Fig. 6). The remaining PCs each contribute less than 10% of the total variation. A PCA of the species means (not shown) produced equivalent results (PC1 = 42.6%, PC2 = 33.4%, PC3 = 11.3%, PC4 = 5.9%), and with the same variables contributing highly on each axis, and thus we present only one analysis for brevity.

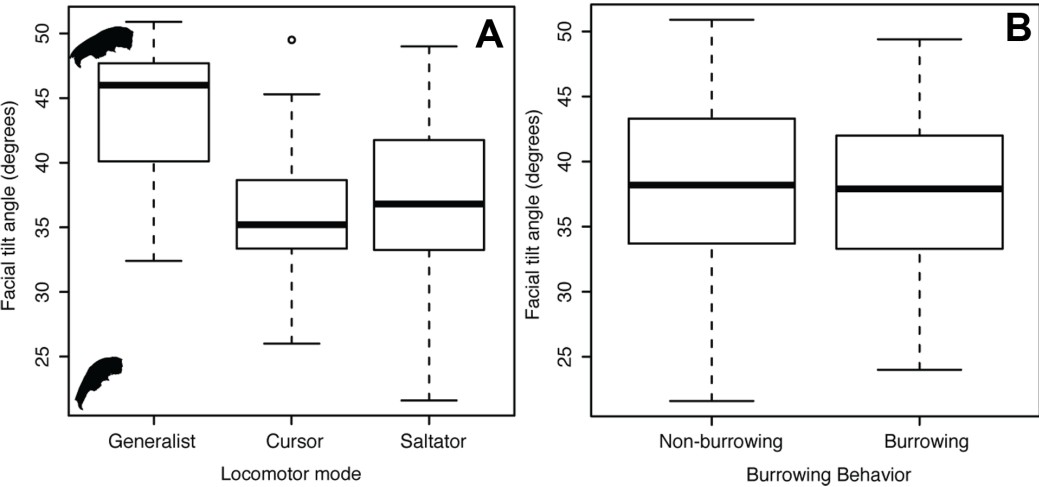

**Figure 5 Facial tilt ANOVA.** Box and whisker plot summarizing facial tilt angle for all specimens, showing how the angle differs between locomotor types (A) and burrowing behavior (B).

The loadings of the PCA (Table 3) show that bullae diameter (BD; 0.85) has the strongest influence on PC1, substantially more than other variables. PC1 strongly separates *Brachylagus*, *Romerolagus*, and *Bunolagus* (all with larger bullae diameters) from all other leporid species (Fig. 6A). In terms of locomotor styles, cursorial species are isolated towards the negative portion of the PC1 axis. Similarly, PC2 strongly shows the effects of size of the *splenius capitus* fossa (SCF; −0.91). While this measure does not separate among saltators (Fig. 6B), there is some separation between generalists and cursorial species. Loadings for PC3 indicate that three variables are strongly affecting the variance along that axis: interorbital width (IOW; 0.68), nasal length (NL; −0.51), and diastema length (DILU; −0.42). PC3 shows separation of species (Fig. 6C), but no clear broader groupings. It also does not clearly distinguish locomotor modes, but saltators do occupy the negative portion of the axes where no generalists or cursors are found (Fig. 6D). Parietal length (PAL; −0.87) loads strongly along PC4. This axes does help to distinguish species (Fig. 6C), but shows little ability to distinguish among locomotor modes (Fig. 6D).

There is significant phylogenetic signal is cranial shape described by the ten log-shape ratios ($K_{mult} = 0.91$, $P = 0.035$). The value of kappa is substantially higher than that for facial tilt, but still below 1, implying that close relatives are moderately less similar to one another phenotypically than expected under Brownian motion. Phylogenetically informed analysis of variance (D-PGLS) indicates that there is no significant effect of locomotor habit on cranial shape ($F = 1.3712$, $P = 0.28$). Likewise, there is no significant effect on cranial shape by burrowing behavior ($F = 1.2831$, $P = 0.56$). Finally, a phylogenetically informed multivariate regression suggests that facial tilt angle is not a significant predictor of cranial shape ($R^2 = 0.097$, $P = 0.413$)

## DISCUSSION

Given a clear correlation between the degree of facial tilt (FT) and locomotor style, and the lack of significant phylogenetic signal in FT angle, it is evident that this aspect of

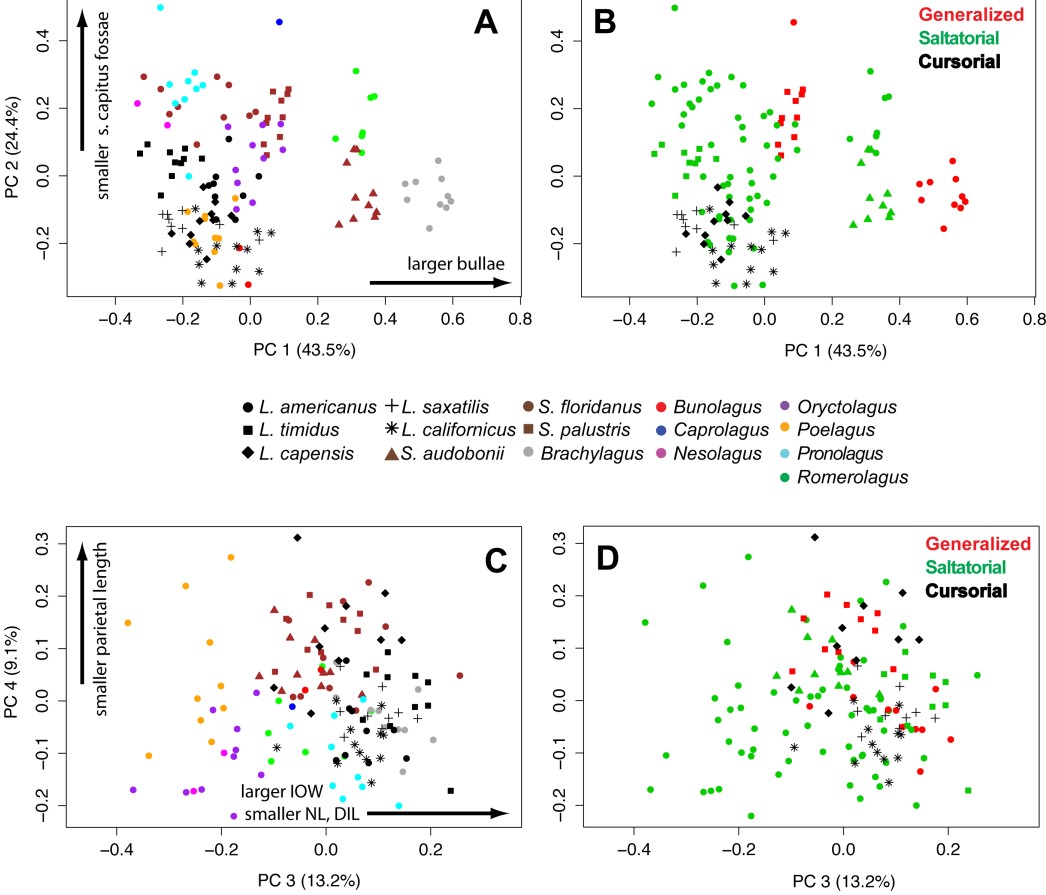

**Figure 6 Multivariate PCAs.** Principal components analysis of 10 log-shape ratios measurements describing cranial shape for all specimens. Biplots show PC1 vs PC2 (top) and PC3 vs PC4 (bottom). (A) Colored symbols by species. (B) colored by LOC (with species symbols). Details of the loadings of each variable in the PCA are presented in Table 3.

cranial morphology is strongly influenced by ecological factors within Leporidae. While the relationship between shape and function is established, the specific aspects of cranial shape that inform ecological function are only partially resolved from our multivariate analyses. Generalized locomotors exhibit less facial tilt, an anatomical condition that could properly be thought to be primitive for the mammalian skull, and given their fossil record, lagomorphs as well (*Dice, 1933*; *Asher et al., 2005*). Facial tilt within leporids is allowed via the expansion of the supraoccipital bone on the dorsal skull (Fig. 2), and along the ventral skull, there is a pronounced flexure near the basisphenoid/presphenoid juncture.

The complex architecture of the supraoccipital in leporids is the most marked change related to the dorsal arching the skull roof, but there are additional effects on the orientation of the orbit (Fig. 2). There is a vast literature on orbital orientation as it relates to locomotion, visual acuity, brain size, and masticatory anatomy (*Noble, Kowalski & Ravosa, 2000*; *Heesy, 2005*; *Heesy, Ross & Demes, 2007*; *Iwaniuk et al., 2008*; *Heesy, 2008*; *Cox, 2008*; *Jeffery & Cox, 2010*), and perhaps most clearly, changes in orbit orientation

have direct affects on the range of visual fields. Both orbital convergence and frontation are commonly measured orbital variables that seem to be functionally predictive (*Cox, 2008*); increased convergence is thought to increase binocular field overlap within primates (*Ross & Martin, 2007*; *Heesy, Ross & Demes, 2007*), and orbital frontation allows for better substrate visualization (*Cartmill, 1970*; *Heesy, 2008*).

While *Jeffery & Cox (2010)* demonstrated that the rabbit shows low degrees convergence and frontation, frontation in the rabbit is complicated by skull transformations associated with facial tilt. Traditionally, frontation was considered as the degree to which the orbital plane is aligned vertically (*Cartmill, 1970*; *Ross, 1995*); whereas, *Jeffery & Cox (2010)* used angular differences between the lateral semicircular canal and the medial and lateral orbital rectus muscles as a proxy for frontation. While these later measures are distinct for rabbits (*Cox, 2008*; *Jeffery & Cox, 2010*) as compared to other mammals, due to the way rabbits hold their heads (*De Beer, 1947*; *Vidal, Graf & Berthoz, 1986*), angular differences between the lateral semi-circular canal and horizontal rectus muscles may not be a perfect summary of the degree to which the orbital plane approaches vertical. Interestingly, as *Jeffery & Cox (2010)* show, humans and rabbits are outliers with regard to this metric, as they both demonstrate strong misalignment of semicircular canal and rectus muscle orientations. This may be driven by the fact that both of these species exhibit skull shapes in which the basicranium is highly transformed relative to the facial region (*DuBrul, 1950*). Most importantly, and regardless of the absolute degree of frontation, facial tilting within leporids would have the effect of changing the orientation of the orbit and increasing frontation. Our data show that variation in facial tilt among leporids (∼30°) is explained by mode of locomotion. Presumably, pronounced facial tilt and the associated increase in frontation improve substrate visibility in fast-moving taxa.

In contrast to FT angle, overall cranial morphology as described by ten log-shape ratio measurements is not significantly different among locomotor modes or between burrowers and non-burrowers. Instead, the PCA of individual variation in our cranial variables clearly shows that among-species variation is a strong driver of morphospace organization (Fig. 6). The phylogenetic structure evident in the cranial variables shown in our PCA is supported by a high measure of phylogenetic signal. However, there is some separation of the three locomotor modes in morphospace (Fig. 6). Saltatorial species have a wide-variety of cranial morphologies, while the generalized locomotors are clustered in morphospace (in the negative quadrant of PC1 and PC2), likely due to their close ancestry.

Bulla length contributes the most to the first PCA axis, separating out a group of three species (larger bullae; *Romerolagus*, *Bunolagus*, and *Brachylagus*) from all other leporids, and thus this morphological trait is a candidate feature of adaptive differences between the different locomotors styles. The external bulla is a complicated structure, which receives contributions from different bones across Mammalia (*Novacek, 1977*). The external auditory bulla has been shown to be of significant systematic importance within carnivorans (*Hunt, 1974*; *Ivanoff, 2001*), but the function of bulla size is unclear for leporids. *Pavlinov & Rogovin (2000)* showed that bulla size is negatively correlated with pinna size in specialized desert rodents. They specifically remark that faster, more agile

rodents within these groups tend toward smaller bulla and larger pinnae. While our data do not explicitly test this, our cursorial species appear toward the negative PC1 axes, which are represented by smaller bullae. We also note that while *Romerolagus* and *Brachylagus* exhibit large bullae and relatively small pinnae, *Bunolagus* does not fit this pattern as it has both large bullae and large pinnae. *Liao, Zhang & Liu (2007)* shows that bulla size is negatively correlated with altitude in the Daurian pika, *Ochotona dauurica* (Lagomorpha, Ochotonidae). This patterns does not match our observations, as one of our large bullae species, *Romerolagus*, is found at high elevations (*Cervantes, Lorenzo & Hoffmann, 1990*). We think that our PC1 axis may also reflect the relative size of the basicranium to the facial region within leporids, in addition to bullae size; we discuss this topic further below.

The second variable of interest, which loads strongly on PC2, is the *splenius capitus* fossae. Lateral to the external occipital protuberance (EOP; i.e., Inion) are two large fossae that extend to the parietal/occipital suture and allow for attachment of the *splenius capitus* mm. (*Barone, Pavaux & Blin, 1973*), which are involved in head extension and lateral rotation. The fossae can be clearly identified via the prominent superior nuchal line that extends rostrally from the EOP. The *longissimus capitus* m. inserts with the *splenius capitus* m. in the lateral, mastoid area, of the occipital region. A final long extensor muscle, the *semispinalus capitus* m., attaches to the lateral portions of the EOP. Together, these three long erectors serve to extend, stabilize, and laterally rotate the head (*Igarashi et al., 2000*). Upon comparison of leporid skulls (Fig. 1), it is apparent that those with significant facial tilt are expanding the rostral portions of the supraoccipital bone relative to the caudal portion, and indeed, this seems to be reflected in PC2 as the variance along that axis helps to separate out cursors (larger *splenius capitus* fossa) and generalists (smaller *splenius capitus* fossa). The expansion of the *splenius capitus* fossa should serve to increase the attachment area for the long extensor muscles, allowing for improved extension and lateral rotation of the head.

It is worth noting that all variables strongly affecting PC1, PC2, and PC4 are associated with the neurocranium, and variables affecting PC3 are all associated with the splanchnocranium. It has been thoroughly demonstrated within our own lineage, and mammals more broadly, that these basicranial and facial regions demonstrate strong levels of phenotypic independence (see, for example, *Porto et al., 2009*; *Drake & Klingenberg, 2010*; *Sanger et al., 2012*; *Klingenberg, 2013*) While this pattern is debated within humans and other great apes (*Singh et al., 2012*; *Mitteroecker et al., 2012*; *Martínez-Abadías et al., 2012*), some similarities in skull transformation between humans and rabbits have been noted in the literature (*DuBrul, 1950*; *Moore & Spence, 1969*). *Moore & Spence (1969)* highlighted that both humans and rabbits transform the facial regions relative to the basicranium, but also pointed out that the transformation seems to be driven in the facial regions within rabbits, whereas it seems to be driven from the basicranium in humans. While our data do not directly address modularity or developmental pathways within leporid skulls, it would be useful to understand how relative transformation between the facial and basicranial regions within leporids, which seems to influence facial tilt, could be explained mechanistically by these developmental trajectories.

Most importantly, it is striking that facial tilt *does* distinguish generalist locomotors clearly from more active taxa within leporids. This suggests that FT represents a meaningful biological metric among leporids, but may also summarize a specific aspect of cranial shape not recognized, but alluded to, within our linear variables. While our linear measurements failed to strongly discern differences among locomotor groups, this may be a function of the limited ability of these variables to capture important shape differences among crania within leporids due to the highly transformed nature of their skulls (e.g., pronounced dorsal arching). Nonetheless, our linear variables do separate taxonomic groups, as has been done in other studies (see, for example, *Palacios et al., 2008*; *Pintur et al., 2014*).

Our study demonstrates that the dorsal arching found within leporid skulls, mainly represented here as facial tilt, has a strong relationship with how these animals moved. Facial tilt is related to a complex transformation of nearly all aspects of the leporid skull, including basicranial rearrangement and facial changes in the diastema region. Our linear variables, in addition to distinguishing taxonomic groups, also capture some aspects of these changes related to locomotion. Based on the changes in orbit orientation that are associated with increased facial tilt, it is likely that skull transformations in crown leporids are driven by a need for increased visual perception of substrate.

## ACKNOWLEDGEMENTS

We are grateful to Neil Duncan and Eileen Westwig of the American Museum of Natural History and Jim Dines of the Natural History Museum of Los Angeles County for access to specimens in their care. We thank Christopher Heesy and Kevin Middleton for helpful discussions, and Margaret Metz for assistance with R.

### Funding

This project was initiated under a fellowship from the AMNH awarded to Brian P. Kraatz. Nicholas Brumacod completed work on this project while enrolled in Western University of Health Sciences' MSMS program, and received a summer fellowship from the Graduate School at that university to continue that work. Travel to the AMNH by Matthew J. Wedel was made possibly by funds from the Department of Anatomy, Western University of Health Sciences. The funders had no role in study design, data collection and analysis, decision to publish, or preparation of the manuscript.

### Grant Disclosures

The following grant information was disclosed by the authors:
AMNH.
Western University of Health Sciences' MSMS.
Department of Anatomy, Western University of Health Sciences.

## Competing Interests

Brian P. Kraatz and Mathew J. Wedel are Academic Editors for PeerJ.

## Author Contributions

- Brian P. Kraatz conceived and designed the experiments, performed the experiments, analyzed the data, contributed reagents/materials/analysis tools, wrote the paper, prepared figures and/or tables, reviewed drafts of the paper.
- Emma Sherratt performed the experiments, analyzed the data, contributed reagents/materials/analysis tools, wrote the paper, prepared figures and/or tables, reviewed drafts of the paper.
- Nicholas Bumacod performed the experiments, analyzed the data, contributed reagents/materials/analysis tools, wrote the paper, reviewed drafts of the paper.
- Mathew J. Wedel conceived and designed the experiments, contributed reagents/materials/analysis tools, wrote the paper, prepared figures and/or tables, reviewed drafts of the paper.

## Supplemental Information

Supplemental information for this article can be found online at http://dx.doi.org/10.7717/peerj.844#supplemental-information.

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
