# Peer review of "Ecological correlates to cranial morphology in Leporids (Mammalia, Lagomorpha)"

_PeerJ, doi:10.7717/peerj.844_

## Round 0.1 · original submission · Major Revisions

I concur with the reviewers in that the study by Kraatz and collaborators is an interesting contribution to evolutionary morphology. The study is quite straightforward in its current form, yet I concur as well in that the study needs several analytical additions. While the phylogenetic sampling is reasonable, It would be desirable to see whether the phylogenetic structure of the data might actually be underlying the observed pattterns (or to what extent). In my view, however, it is suprising that the authors have used traditional morphometrics instead of geometric morphometrics, the latter which would be the most robust way to test their hypotheses. In effect, given the fact that the authors have the original (digital) pictures, implementing these methods would not require much additional work, while greatly enhancing the study. Moreover, as one of the reviewers suggests, the correlation between skull shape and the FT can statistically tested e.g., with multivariate regression, and graphically shown in a more self-explanatory way.
If the authors choose not to apply GM, I agreee in that it would be essential to test the relationships between the variables, say, on a PCA (based on the correlation matrix, given that there would be lengths and angles altogether). Ultimately, there are many suggestions that would help to improve this study, including rewriting the introducction and the discussion (plus adding more references). It's a choice of the authors' to justify which of these additions can help to improve this study.

Reviewer 1 ·

Basic reporting

After reading the revision of the paper of Kraatz and collaborators entitled "Ecological correlates to cranial morphology in Leporids (Mammalia, Lagomorpha)"), I think that this study contains valuable work on the Lagomorpha morphology. Nevertheless, I have a number of comments/modifications to suggest:
1- My first major point is the lack of background literature in this study. The authors need to support their claims (absence of literature in the introduction and discussion that I will discuss below, part by part)
2- My second comment is about the treatment of the linear measurements, I wondered why the authors did not use log shape ratios (Mosimann and James, 1979) in order to study the shape with their linear measurements? In their study they often write that they study the shape of the skull which is not the case (they study the morphology). Furthermore, I am not sure whether the authors log-transformed the linear measurements. (Mosimann JE, James FC. 1979. New statistical methods for allometry with application to Florida redwinged blackbirds. Evolution 33:444-459.)
3- My third main point is that I would like to know if there is convergence of lifestyle in Lagomorpha? I would be happy to see a figure of the phylogeny with the lifestyle plotted on it to see the distribution of the behavior along the phylogeny. Then depending on whether there is convergence of locomotor behaviours (or not) in Lagomorpha, authors will be able to discuss the fact that the morphology of the skull of Lagomorpha is due to phylogeny and/or lifestyle and so one. Furthermore, it would be important for the authors to calculate the phylogenetic signal in their data.

Experimental design

There is the lack of background literature in this study. Furthermore, the authors can calculate the phylogenetic signal for their linear measurement even if their sample is quite low. They can also perform log shape ratio on their linear measurements (Mosimann and James, 1979)
Authors need to describe their results (for example: PCA and the distribution of the species on it as well as the loadings....)

Validity of the findings

Authors need to further describe their results and also to assess what they claim with literature. Especially the discussion is the weakest part of the paper.

Comments for the author

INTRODUCTION

- Line 37 add references after vertebrate groups
- Line 44 add references at the end of the sentence
- Line 45 I do not totally agree with the fact that the skull is more often overlooked in studies of form and locomotion, authors need to talk about the fact that the skull is a complex structure (see Hanken & Hall, 1993) and is influenced by several factors due to functions involved such as protection of the brain, smell, eyesight, breathing and mastication (Marroig & Cheverud 2001, 2004). In the literature, the skull is more often studied in studies of form and feeding and in some case locomotion as the interesting example that they give.
-
Hanken J, Hall BK 1993. The skull. Volume 3. Functional and evolutionary mechanisms. Chicago, IL: University of Chicago Press.
Marroig G, Cheverud J-M. 2001. A comparison of phenotypic variation and covariation patterns and the role of phylogeny, ecology and ontogeny during cranial evolution of New World monkeys. Evolution 55: 2576–2600.
Marroig G, Cheverud J-M. 2004. Did natural selection or genetic drift produce the cranial diversification of neotropical monkeys? The American Naturalist 163: 417–428
For this second paragraph, it would be nice to add a figure with all the anatomical structures that the authors study, it can help the reader to understand the specialized anatomy of the lagomorpha (Figure 1 is not enough, or alternatively the authors need to add some arrows with the name of the different structures).

- Line 64 I would change locomotion by locomotor behavior or lifestyle

- Line 66 the authors said that they use 17 extant taxa whereas they said 16 in the material and methods (l 127)


- Line 63 and 67 change shape by morphology (except if you decide to use log shape ratio)

- Line 70 remove “today”

- Sentence from line 76 until line79: I am not sure of the relevance of this sentence for the study.

- Line 80, what do the authors mean by “biologically conserved”? and concerning the ecological diversity I wondered if the locomotor strategies originated several times independently in Lagomorpha? And especially in the sample of the authors. Adding a figure with the phylogeny of the species used in the study and the locomotor behaviors and fossorial ability would be important and needed to illustrate that.

- Line 80 to 94, authors really need to add some literature

- Line 92 capitalize the letter “t” of “table 1”

- Line 101-102 concerning the literature, the authors really need to read and integrate the following literature (especially the last one Cox (2008)) which is a review and can highly improve the paper (especially the discussion part of the paper):

Heesy CP, Ross CF & Demes B (2007) Oculomotor stability and the functions of the postorbital bar and septum. In: Primate origins: Adaptations and Evolution. (Eds Ravosa MJ, Dagosto M) Pp.257-283. Springer US
Heesy CP (2005) Function of the mammalian postorbital bar. J Morphol 264:363-380
Heesy CP (2008) Ecomorphology of orbit orientation and the adaptive significance of binocular vision in primates and other mammals. Brain, Behavior and Evolution 71:54-67
Iwaniuk AN, Heesy CP, Hall MI, Wylie DRW (2008) Relative wulst volume is correlated with orbit orientation and binocular visual field in birds. J Comp Physiol A 194:267-282
Noble VE, Kowalski MK, Ravosa MJ (2000) orbit orientation and the function of the mammalian postorbital bar. J Zool. Lond 250, 405-418
Cox PG, Jeffery NJ (2008) Morphology of the mammalian vestibule ocular reflex: the spatial arrangement of the human fetal semicircular canals and extraocular muscles. J morphol 268:878-890.
Jeffery NJ, Cox PG (2010) Do agility and skull architecture influence the geometry of the mammalian vestibule-ocular reflex? J Anat 216:496-509.
Lieberman DE, Ross CF, Ravosa MJ (2000) Primate Cranial base: Ontogeny, function and integration. Yearbook of physical anthropology 43:117-169.
Cox PG (2008) A quantitative analysis of the Eutherian orbit: correlation with masticatory apparatus. Biol Rev 83:35-69.

- Line 110 replace workers by authors, again the authors need to add some literature to this paragraph.

MATERIAL AND METHODS
The authors can calculate the phylogenetic signal for their linear measurement even if their sample is quite low.
They can also perform log shape ratio on their linear measurements (Mosimann and James, 1979)
Mosimann JE, James FC. 1979. New statistical methods for allometry with application to Florida redwinged blackbirds. Evolution 33:444-459.
- line 174 and 187 replace R Development Core Team 2014 by R Core Team 2014
http://cran.r-project.org/ then go to “frequently asked questions” and “R Faq” and “2.8 Citing R”
To cite R in publications, use
@Manual{,
title = {R: A Language and Environment for Statistical
Computing},
author = {{R Core Team}},
organization = {R Foundation for Statistical Computing},
address = {Vienna, Austria},
year = 2014,
url = {http://www.R-project.org}
}

RESULTS

Authors need to italicize the ‘F’ and ‘P’ and to present their result in order to understand which results takes into account the phylogeny and which not. Both kinds of results need to be presented in order to understand whether the phylogeny has an effect on the morphology of skull.
Authors need to describe the PCA and the distribution of the species on it as well as the loadings. They really need to go further in their results. That will help them to go further in their following discussion.
They need to present the scatterplot for the axis 2/3 in the figure 4 (and by the way they need to add the percentage of variance explained in the axis legends of the scatterplot). It would be helpful to do a PCA on the mean of species and plot the phylogeny on this PCA.

DISCUSSION

This is the weakest part of the paper. The authors need to read some papers on the topic in order to interpret and discuss their results (already the publications cited above can help them) and also to assess what they claim ( there are only 4 publications cited in their discussion…). Furthermore, if they can do some further analyses or interpretations of their results, I think they will be able to discuss better discuss their results. For example if some (M)AN(C)OVA are significant without taking into account the phylogeny in comparison of those that are significant taking into account the phylogeny, it would mean that phylogeny has a big effect on the morphology of the skull of leporids and so on.

Figure 3: nice figure and the outline of the rabbit tilt makes it easy to read. Authors just need to add the “a” and “b” on the figure (it appears only in the legend of the figure).

Figure 4 change “LOC” in the legend of the figure by locomotor behavior of locomotion… change the labels of the scatterplot by adding the percentage of variance and add the scatterplot 2/3

Table 1. Authors can remove the geographic range and the abbreviation that they do not use in the study.

Table 3 the legend need to be a sentence not only “PCA loadings”

Reviewer 2 ·

Basic reporting

No comments

Experimental design

No comments

Validity of the findings

No comments

Comments for the author

The paper titled “Ecological correlates to cranial morphology in Leporids (Mammalia, Lagomorpha)” of Kraatz and colleagues tests for association in morphology and size towards locomotion strategy in the skull of crown leporids. The authors use a plethora of multivariate methods such as MANCOVA, PCA or PGLS to explore the association between both cranial shape and facial tilt (FT) with locomotion style, under a well-stablished phylogenetic framework.
I personally find very interesting both hypotheses tested in the MS. The methodology seems to be well applied. The discussion and interpretations represent a welcome addition to the field of mammalian ecomorphology; it is surprising that changes in the cranial skeleton reflect locomotor adaptations when usually solely reflect dietary adaptations. For this reason, I recommend the publication of the paper, although I have detected some problems that, in my opinion, need to be solved prior publication. Below, I enumerate some suggestions and minor points that need clarification.
1. Given that the authors have digital pictures of the specimens in lateral view, I would suggest the use of geometric morphometrics to explore if the main axes of cranial shape variation are associated with FT variation. I think that the paper would be highly improved using such an approach as it could support the results obtained.
2. I also suggest to perform a canonical variates analysis to explore the cranial shape features that distinguish the three modes of locomotion styles. With such an approach the authors could document if those cursorial and saltatorial taxa differ from generalized taxa in FT or any other cranial trait. If you have more variables than cases per group, you can use a between-group PCA (see Mitteroecker and Bookstein 2011; Evol. Biol. 38:100–114). If the authors decide to maintain their traditional methods, instead of the geometric methods that I suggest, I would also recommend to include the FT as a variable together with the other metric variables used.
3. On their “Hypothesis-1”. If they use the FT as a proxy for frontation (L.113), then, why they do not measure directly the frontation angle? As there is vast literature dealing on this angle, I recommend to measure the frontation angle directly on the skulls or at least to give a compelling reason on the use of the FT as a proxy. This will clarify this point to non-specialize readers.
4. The introduction is some vague, as there are several assertions made by the authors that need reference. For example, this is the case of Line81 when the authors state that the crown group of leporids exhibits “a surprising level of ecological and morphological diversity”. There are more cases like this in the introduction that I have mentioned in the minor comments (see below). In my opinion the introduction should be rewritten joining the information given under the sections of “Introduction” and “Study System and Hypothesis”.

Minor problems:
Introduction. L. 37. Please give examples.
Introduction. L.40-41. Please revise the definition of “cursoriality” as it relates to more than “high-speed”.
Introduction. L.53-56. It is hard to follow the anatomical terminology of the skull mentioned in this paragraph. I think that this information should be indicated in Fig. 1.
Study System and Hypothesis. L. 81. Please, include a reference. ”Despite the perception that leporids are biologically conserved, the crown group exhibits a surprising level of ecological and morphological diversity”.

Study System and Hypothesis. L. 82. Please, include a reference. “Leporids are found on every continent except Antarctica, from the high arctic to dry, hot deserts”

Study System and Hypothesis. L. 85. Please, include a reference. “Some leporids are nocturnal, some are social, and some live in dense cover as opposed to the open plains often associated….”
Study System and Hypothesis. L. 88. Please, include a reference. “These cranial differences manifest themselves morphometrically via a wide range of snout lengths and marked differences in skull robustness and form”
Hypothesis 1—Facial Tilt. L. 102. Please substitute “ecomorphology” by “ecology”.
Hypothesis 1—Facial Tilt. L. 105. Ross, 1993 is not in the reference list.
Hypothesis 1—Facial Tilt. L. 113. How do you know that leporids have a relatively low degree of convergence but high degree of frontation? It is just a visual impression? Do you have a reference? Have you tested this statistically? Please include an argument, otherwise this statement is vague, or at least, subjective.
Why the authors think that skull shape might change with locomotor mode or burrowing habit in the “Hypothesis-2”? Is there a specific reason? Besides, the authors state that Hypothesis-2 is therefore more a form of exploratory data analysis than a test for specific hypothesis” (L. 124). Why is then treated as a hypothesis?
Material and Methods. Line 131. Some of the metric variables used by the authors are highly redundant and this could bias the variance/covariance matrix calculated to perform PCA. This is particularly true for SLD and SLV or NW and IOW. A solution could be to combine them (e.g., to substitute SLV by a new variable calculated as SLV minus SLD) and reanalyze the data, just to see if the analysis change. At least this should be mentioned in the paper.
Material and Methods. Line 132. Please give some detail on the study of repeatability used.
Material and Methods. Line 145-151. Please, include the software used to assemble (or to prune) the tree.
Results. Lines 196-199. It would be interesting to add the same plot of Fig. 3 but with M1A as a control for body mass. In this way, readers could see if the differences observed between generalists from cursors plus saltators are not just because generalist species are bigger or smaller than the other two groups. I know that the effect of body size on locomotor style has been tested but I think that it is a good idea to show this plot.
Results. Lines 202-205. What happens if the PCA is computed from species averages instead of specimens? What happens if the PCA is computed including the facial tilt as a variable? What happens if the PCA is computed working with variables free of evolutionary allometric effects (e.g. with Mossimann variables)? I think that this should be at least explored.
Results. Lines 206-212. I do not understand why the authors do not test the effect of the facial tilt on cranial shape (if the former could be decomposed from the second).
Discussion. Lines 220-221. Is it similar than in dog breeds? (see Schoenebeck et al. 2012; Plos genetics).
Discussion. Lines 250-252. Yes! This is what I suspected because I am not sure if it is possible to decompose the facial tilt from skull shape, as it seems that a significant part of cranial shape variation is due to changes in the facial tilt. For this reason, I think that a good approach to follow is to work with geometric morphometrics as I am sure that the first PC will account for differences in the facial tilt among the sample.
Discussion. Lines 255. Please, substitute the word “discriminate” by “separate” as discriminant analysis is not used. Furthermore, the authors say that the first eigenvector reflects size and the second reflects shape but they do not explain why. I understand that this is deduced from the factor loadings of Table 2 but I would specify this to avoid confusion to readers.
I think that PCA should be commented more deeply. The variables loading on both eigenvectors, etc.

---

## Round 0.2 · accepted · Accept

The authors have thoroughly revised their first version of the manuscript following the suggestions by the reviewers in detail; in my opinion the study is a highly improved and interesting contribution to vertebrate morphology.

Please note that I have found what seemed inconsistencies in the text, between Figure citations and their corresponding numbers (see my guesses below). Please track this in the proofs (if the mismatch holds) during the publication process.

Line 116; reads Fig. 3, should be Fig 2
Line 153; reads Fig. 3, should be Fig. 4
Line 229; reads Fig. 3, should be Fig. 2
Line 325; reads Novaek, 1977, should be Novacek, 1977